# Storage Fungi and Mycotoxins Associated with Rice Samples Commercialized in Thailand

**DOI:** 10.3390/foods12030487

**Published:** 2023-01-20

**Authors:** Seavchou Laut, Saranya Poapolathep, Onuma Piasai, Sujinda Sommai, Nattawut Boonyuen, Mario Giorgi, Zhaowei Zhang, Johanna Fink-Gremmels, Amnart Poapolathep

**Affiliations:** 1Department of Pharmacology, Faculty of Veterinary Medicine, Kasetsart University, Bangkok 10900, Thailand; 2Department of Plant Pathology, Faculty of Agriculture, Kasetsart University, Bangkok 10900, Thailand; 3Plant Microbe Interaction Research Team (APMT), National Center for Genetic Engineering and Biotechnology (BIOTEC), National Science and Technology Development Agency (NSTDA), Khlong Nueng, Khlong Luang, Pathum Thani 12120, Thailand; 4Department of Veterinary Science, University of Pisa, 56124 Pisa, Italy; 5Oil Crops Research Institute of the Chinese Academy of Agricultural Sciences, Wuhan 430062, China; 6Institute for Risk Assessment Sciences, Faculty of Veterinary Medicine, Utrecht University, 3584 Utrecht, The Netherlands

**Keywords:** rice, *Aspergillus*, *Penicillium*, *Talaromyces*, mycotoxins, fungal systematics, LC-MS/MS

## Abstract

The study focused on the examination of the different fungal species isolated from commercial rice samples, applying conventional culture techniques, as well as different molecular and phylogenic analyses to confirm phenotypic identification. Additionally, the mycotoxin production and contamination were analyzed using validated liquid chromatography–tandem mass spectrometry (LC-MS/MS). In total, 40 rice samples were obtained covering rice berry, red jasmine rice, brown rice, germinated brown rice, and white rice. The blotting paper technique applied on the 5 different types of rice samples detected 4285 seed-borne fungal infections (26.8%) for 16,000 rice grains. Gross morphological data revealed that 19 fungal isolates belonged to the genera *Penicillium/Talaromyces* (18 of 90 isolates; 20%) and *Aspergillus* (72 of 90 isolates; 80%). To check their morphologies, molecular data (fungal sequence-based BLAST results and a phylogenetic tree of the combined ITS, *BenA*, *CaM*, and *RPB2* datasets) confirmed the initial classification. The phylogenic analysis revealed that eight isolates belonged to *P. citrinum* and, additionally, one isolate each belonged to *P. chermesinum, A. niger*, *A. fumigatus*, and *A. tubingensis.* Furthermore, four isolates of *T. pinophilus* and one isolate of each taxon were identified as *Talaromyces* (*T. radicus*, *T. purpureogenum*, and *T. islandicus*). The results showed that *A. niger* and *T. pinophilus* were two commonly occurring fungal species in rice samples. After subculturing, ochratoxin A (OTA), generated by *T. pinophilus* code W3-04, was discovered using LC-MS/MS. In addition, the *Fusarium* toxin beauvericin was detected in one of the samples. Aflatoxin B1 or other mycotoxins, such as citrinin, trichothecenes, and fumonisins, were detected. These preliminary findings should provide valuable guidance for hazard analysis critical control point concepts used by commercial food suppliers, including the analysis of multiple mycotoxins. Based on the current findings, mycotoxin analyses should focus on *A. niger* toxins, including OTA and metabolites of *T. pinophilus* (recently considered a producer of emerging mycotoxins) to exclude health hazards related to the traditionally high consumption of rice by Thai people.

## 1. Introduction

Rice belongs to the genus *Oryza* and the family Poaceae and is recognized as a staple food that provides up to 50% of the dietary caloric supply as well as a substantial part of the protein intake and important nutrients for the population in the Asia-Pacific region [1]. Most of the rice consumed is *Oryza sativa* (known as Asian rice), and *O. glaberrima* Steud. (Steudel) [2]. In Thailand, rice is also the main food and source of nutrition for most citizens, with rice production being the most important food crop as well as a major component of the Thai economy [3]. Thailand was the world’s sixth largest rice producer in 2018, with a total rice production of 32.2 million metric tons (FAOSTAT, 2020; https://www.fao.org/faostat/en/#home, accessed on 21 November 2022). However, several conditions, including the unsuitability of the plant hybrid to the local environment, drought, insect infestation, or destructive mechanical harvesting, can magnify mycotoxin production in the field, while high temperature, moisture, and pest incidence magnify mycotoxin accumulation under storage conditions [4]. Cereal grains, such as rice, are easily contaminated by fungal species, with rice commonly cultivated using flood irrigation, where high levels of moisture favor fungal proliferation and mycotoxin production [5]. In addition, poor harvesting techniques can increase the vulnerability of the harvested crop to fungal invasion during transport, processing, and storage [6].

Mycotoxins are known as toxic secondary metabolites of numerous fungal species and represent a diverse group of small chemical substances, which persist in food commodities during storage and processing [7]. The majority of mycotoxigenic fungal species belong to the genera *Aspergillus, Fusarium, Talaromyces*, and *Penicillium*, which can synthesize several well-described mycotoxins [8,9]. Fungal invasion can occur during plant growth in a contaminated environment (pre-harvest contamination) as well as during transport and storage (post-harvest contamination) [8,10]. Typically, rice grains will be harvested with a rather high moisture content, also facilitating fungal invasion and proliferation in the storage system, resulting in additional invasion during typical storage.

Human health hazards associated with food items contaminated with fungi and fungal conidia (spores) include allergic reactions (particularly following inhalation of spores) and long-term skin contact (occupational exposure), as well as exposure to mycotoxins. Mycotoxins are small molecules with diverse chemical structures and generally very stable during food processing including common kitchen procedures, such as boiling rice. Analysis of mycotoxins in foods requires sophisticated and validated chemical methods.

The major health concerns focus on the mutagenic and genotoxic mycotoxins involved in the pathogenesis of cancer [7]. Exposure to aflatoxin B1 (AFB1) has been experimentally and epidemiologically related to an increase in the prevalence of liver cancer (human hepatocellular carcinoma) and is classified as a human carcinogen (Group 1) by the International Agency for Research on Cancer [11]. Ochratoxin A (OTA) has been identified a potential carcinogen and associated with the prevalence of cancers of the kidney and the urinary tract [12]. Both toxins are regulated by most countries around the world to reduce exposure rates in the human population. Aflatoxins and OTA have also been identified in rice [11] alone and in combination with various other mycotoxins, differing by region and production method. The mycotoxins present in rice are of special concern, as rice is a typical staple food and, as such, contributes substantially to overall human exposure. More than 400 toxic mycotoxins have been recorded in grains, nuts, fruits, vegetables, beverages, and other agricultural commodities; however, only some of these mycotoxins are considered to induce severe adverse effects on human and animal health, such as aflatoxins, ochratoxins (OTAs), trichothecenes, zearalenone (ZEN), fumonisins (FB1 and FB2), and patulin [13].

Although different mycotoxins in rice have been described in various countries [14,15,16,17,18,19,20], only a few studies have reported on mycotoxin contamination of rice in Thailand [21,22,23]. The present study was undertaken to provide insight into the fungal contamination of rice samples of commercial products in supermarkets in Bangkok, Thailand using morphological and molecular methods, as well as using liquid chromatography–tandem mass spectrometry (LC-MS/MS) as an analytical method for multi-mycotoxin determination.

## 2. Materials and Methods

### 2.1. Rice Collection and Fungal Isolation

In total, 40 samples in package form were obtained from 5 types of rice grain, comprising rice berry (13), red jasmine rice (8), brown rice (7), germinated brown rice (6), and white rice (5). Commercially packed rice grains (1–2 samples of 1 kg processed rice grains from each brand) were randomly collected to obtain a representative sample of commercial products in supermarkets in Bangkok, Thailand. The samples were stored at 4 °C until analysis. Fungal isolation from the rice samples followed the blotting paper method [24]. The rice grain samples were analyzed for the presence of major seed-borne fungal pathogens following the International Rules for Seed Testing 2013 [25]. For each variety, 400 rice grains were analyzed in 16 replications. Blotting papers soaked in sterile distilled water were placed in a glass Petri dish. Then, 25 rice grains were placed on 3 layers of moist blotting paper in each dish, maintaining an equal distance between grains in a concentric manner. The Petri dishes were incubated at 25 °C under a 12 h light and 12 h darkness cycle for 7 days. The percentage incidence of individual pathogens was determined. Most of the associated pathogens were detected by observation of their growth characteristics on the incubated seeds on the blotting paper [26].

The percentage frequencies of occurrence of the various fungal species were calculated using [27]:(1)$$\mathrm{Frequency}\;\mathrm{of}\;\mathrm{occurrence}\;\left(\%\right)=\frac{\mathrm{Number}\;\mathrm{of}\;\mathrm{seeds}\;\mathrm{on}\;\mathrm{which}\;a\;\mathrm{fungal}\;\mathrm{species}\;\mathrm{occurs}\times 100}{\mathrm{Total}\;\mathrm{number}\;\mathrm{of}\;\mathrm{seeds}}$$

### 2.2. Morphological Study

Fungi that had grown from the grains on the blotting paper were examined and identified according to colony characteristics and the morphology of their sporulation structures under a light microscope. Notably, in most cases, more than one type of fungal colony was observed using the agar plate method [28]. All isolates were subcultured on plates of new potato dextrose agar (PDA, Difco, Detroit, MI, USA) for slide culture. The PDA dishes were incubated at 25 °C for 7 days. Temporary slides were prepared from the fungal colony and observed under a light microscope at 40x (Carl Zeiss; Scope.A1; Jena, Germany), according to described methods [26,27,28,29,30]. The fungi detected on the incubated grains were transferred to PDA. The cultures were incubated at 25 °C for 3–7 days. The results were presented as the percentage incidence for individual pathogens [28].

### 2.3. DNA Extraction, Polymerase Chain Reaction Amplification, and Sequence-Based Identification

Total genomic DNA of fungal mycelia grown on PDA ranging from 25–28 °C for 14 days was extracted using the CTAB method [31]. Identification of the fungal isolates was carried out using polymerase chain reaction (PCR)-based sequences from 4 loci—the internal transcribed spacer (ITS), beta tubulin (*BenA*), calmodulin (*CaM*), and the RNA polymerase II second largest subunit (*RPB2*)—using the following primers: ITS1/ITS4 for the ITS [32,33], 5F2/7CR for *RPB2* [34], CMD5/CMD6/CL1/CL2A for *CaM* [35,36], and BT2A/BT2B for *BenA* [33]. Electrophoresis (1.5% agarose gel) was carried out and visualized using ultraviolet light for the size determination of the PCR products [37]. The PCR amplicons were purified by Macrogen Inc. (Seoul, South Korea), where DNA sequences were determined using the same PCR primer used in the DNA amplification (Table 1). Forward and reverse sequences were trimmed and assembled using the BioEdit software [38].

### 2.4. Multiple Sequence Alignments and Phylogenetic Analyses

The obtained sequences of ITS, *BenA, CaM*, and *RPB2* were assembled and trimmed at both ends using the BioEdit v.7.1.3 software [36]. The newly generated sequences were deposited in GenBank representative taxa as ingroup taxa used in phylogenetic analyses based on fungal datasets [40,41,42], and their accession numbers are provided in Appendix A. *Metarhizium anisopliae* (AF218207, AY995134, and DQ522453) was used as an outgroup species. Multiple sequence alignments were performed separately using the MAFFT v.7.490 software [36] for each locus and adjusted manually. The 4 datasets were concatenated using the BioEdit v.7.1.3 software [36]. Maximum likelihood (ML) phylogenetic analyses, using 1000 bootstrap replicates, were performed using RAxML-NG [38] under the GTR + GAMMA model with default parameters on the Debian Linux operating system. Bayesian posterior probabilities (PPs) were determined using the MrBayes v.3.2.7 software [39] with 5,000,000 Markov chain Monte Carlo generations, with the first 2,000,000 discarded as burn-in. Bootstrapping for maximum parsimony (MP) and maximum likelihood (ML) was performed following the method of Raja et al. (2017) [32]. The consensus tree was visualized and adjusted using the Adobe Photoshop 2021 software and the FigTree v1.4.4 software (http://tree.bio.ed.ac.uk/software/figtree, accessed on 10 September 2022).

### 2.5. Fungal Extraction for Mycotoxin Detection

Mycotoxin production (AFB1, AFB2, AFG1, AFG2, OTA, FB1, FB2, ZEN, deoxynivalenol (DON), citrinin (CIT), nivalenol (NIV), diacetoxyscripenol (DAS), T-2 toxin (T-2), beauvericin (BEA), and alternariol (ALT)) was tested in 9 species: *Aspergillus fumigatus, A. niger, A. tubingensis, Talaromyces radicus, T. purpureogenum, T. islandicus, T. pinophilus, P. citrinum*, and *P. chermesinum.* The analysis was performed using a published method [43] with minor modification. Mycotoxins were extracted from fungal cultures at day 7 of incubation. For each fungal strain, mycelia and conidia were taken from the edge, the center, and midway space of each colony. Mycotoxins were extracted using 2 mL of chloroform-to-formic acid (99:1, *v*/*v*), and vortex mixing for 30 min, followed by sonication for 5 min. Then, the liquid phase was separated using centrifugation at 1968× *g* for 10 min and 500 μL of the supernatant was completely evaporated to dryness under a gentle stream of nitrogen at 40 °C on a heating block. The residue was reconstituted in 500 μL of the mobile phase solution and passed through a 0.22 μm syringe filter (Sartorius AG; Goettingen, Germany) before analysis. The extract was transferred to a vial and analyzed using LC-MS/MS.

### 2.6. QuEChERS Sample Extraction and Clean-Up

The extraction method was performed using a published method [22,44]. Briefly, 1.0 g of rice sample was added with 5 mL of Milli-Q water, followed by 5 mL of 10% (*v*/*v*) acetic acid in acetonitrile (ACN) and mixed for 2–3 min using a vortex mixer. Then, 2.0 g of MgSO_4_, 0.5 g of sodium citrate tribasic dehydrate, and 0.25 g of sodium citrate dibasic sesquihydrate were added into the mixture that was then shaken for 1–2 min. The ACN fraction was separated using centrifugation at 1968× *g* for 5 min. Supernatant fractions of 2 mL were transferred into a tube containing 300 mg of MgSO_4_, 50 mg of C18, and 25 mg of primary secondary amine (PSA), and shaken for 2–3 min. The mixture was separated using centrifugation at 1968× *g* for 15 min. Then, 1 mL of supernatant was completely evaporated to dryness under a nitrogen stream at 40 °C on a heating block. The residue was reconstituted with the mobile phase solution, and passed through a 0.22 μm syringe filter (Sartorius AG; Goettingen, Germany) before being analyzed using LC-MS/MS.

### 2.7. LC-(ESI)-MS/MS Conditions

The liquid chromatography–tandem mass spectrometry (LC-MS/MS) analysis was performed using a method modified from published studies [22,44,45] The chromatographic separation was performed on a ZORBAX Eclipse plus RRHD C18 column (50 × 4.6 mm, 1.8 μm diameter). The temperature in the column was kept constant at 40 °C. The mobile phase consisted of 5 mM ammonium formate with 0.2% formic acid in water (mobile phase A) and 5 mM ammonium formate with 0.2% formic acid in acetonitrile (mobile phase B), with a gradient elution of: 0–1 min, 10% B; 1–5 min, 10–95% B; 5–10 min, 95% B; and 10–12 min, 10% B. The column was re-equilibrated for 3 min between injections. The mobile phase solution was filtered through a 0.22 μm membrane (Sartorius AG; Gottingen, Germany) and ultrasonically degassed prior to application. The analysis used a triple quadrupole mass spectrometer (6460 triple; Agilent Technologies; Waldbronn, Germany) equipped with an electrospray ionization source run in the positive ion mode under the multiple reaction monitoring mode. The ionization source parameters were optimized (Appendix A).

### 2.8. Method Validation Procedure and Matrix Effects Study

Validation of the LC-MS/MS method for multi-mycotoxin (AFB1, AFB2, AFG1, AFG2, OTA, FB1, FB2, ZEN, DON, CIT, NIV, DAS, T-2, BEA, and ALT) detection was performed to assess the efficiency of this analytical method by investigating the recovery, repeatability, linear working range, limit of detection (LOD), limit of quantification (LOQ), accuracy, precision, and matrix effects in accordance with the guidelines of Commission Decision 2002/657/EC [46]. The LOD and LOQ values of the method were evaluated as signal versus noise values of 3:1 and 10:1, respectively (Appendix A). The LOD values for the tested mycotoxins were in the ranges 0.15–25.0 μg/kg, 0.20–28.2 μg/kg, and 0.15–25.6 μg/kg, whereas the LOQ ranges were 0.50–83.33 μg/kg, 0.69–92.8 μg/kg, and 0.51–85.3 μg/kg in the groups of white rice, colored rice, and culture media, respectively. Linear regression analysis was conducted for multiple mycotoxin standards under the optimized LC-MS/MS conditions. Recovery and precision (repeatability, expressed as the relative standard deviation as a percentage) were determined within-day by analyzing 7 replicates of spiked samples at 3 different quality control (QC) levels (Appendix A). The inter-day precisions were determined by analyzing QC samples on 5 different days (one batch per day). The matrix-matched calibration curves were prepared in 3 replicates by spiking the working standard solution into blank samples to yield final concentrations in the specified ranges (Appendix A). The matrix effects were determined by comparing the slope of 6 levels of the calibration curves of the target analyte in post-extraction spiked samples to those obtained in neat solvent.

## 3. Results

### 3.1. Total Seed-Borne Fungal Infections Using Blotting Method

As shown in Table 2, there were 4285 (26.78%) fungal infections detected among the 16,000 grains from the 5 different types of rice (rice berry, red jasmine rice, brown rice, brown rice, and white rice). Table 2 shows that red jasmine rice had the highest level of seed-borne fungal contamination, followed by rice berry (*n* = 1291), germinated brown rice (*n* = 1037), brown rice (*n* = 405), and white rice (*n* = 77), respectively. After preliminary morphological categorization of the fungal infestation of the 4285 infected rice grains, 90 individual samples were further analyzed to accurately determine the individual species.

### 3.2. Preliminary Fungal Identification Using Agar Plate Method

In total, 90 isolates were obtained from 40 rice samples; these fungi that had been cultured on the agar medium were identified based on colony features and fruiting body characteristics. Morphological analysis revealed that 20% (18 of 90 isolates) belonged to the genera *Penicillium* and *Talaromyces,* whereas 80% (72 of 90 isolates) belonged to the genus *Aspergillus* (Table 3). These findings were consistent with other studies carried out in Nigeria, the Republic of Korea, Sri Lanka, Thailand, India, and Turkey, showing that these fungal genera were the most predominantly found in the rice samples [14,19,47,48,49]. In this first step, the phenotypic observations included cultural and morphological characteristics (Figure 1).

### 3.3. DNA Sequence-Based Data of Fungal Isolates Using BLAST NCBI Database Tool

Identification of fungal species based on phenotypic appearance and morphology has been a common technique for many years for the identification of fungi at the ordinal or family level; however, recent molecular techniques allow a reproducible confirmation of a lower-level (species) classification [32,50]. Subsequently, the fungal isolates in the present study were accurately identified using the BLAST tool in the NCBI database as one of the sequence-based methods and compared to the accessible sequences based on four loci (ITS, *BenA*, *CaM,* and *RPB2*) [51]. All 19 fungal isolates produced top hits to sequences from the corresponding species described by the morphological characteristics of the isolates described above (Appendix A). For example, the ITS, *BenA*, *CaM,* and *RPB2* data for isolate G17-01 showed 99.67, 99.81, 100, and 99.89% identities with *Aspergillus niger* based on the accession numbers MH752206, KC175288, MH614646, and MT318291, respectively. The ITS data also gave top hits for isolate B8-03 at 100% (*Talaromyces purpureogenus* KC344977), and for *BenA* (*T. purpureogenus* KC345006) at 99.15% identity. Additionally, DNA sequence comparison of the ITS, *BenA,* and *CaM* loci of isolate G11-05 revealed that the three loci had 100% similarity with *P. chermesinum* (MW081295, KJ767035, and MT302215). In the present study, eight fungal isolates from these rice samples revealed that the predominant filamentous fungal species was *A. niger* (Appendix A). Overall, the genus *Aspergillus* was the most common in the samples examined (72 isolates). The samples yielded 19 isolates in 9 species: *A. tubingensis*, *A. fumigatus*, *A. niger*, *T. radicus*, *T. purpureogenum*, *T. pinophilus*, *T. islandicus*, *P. citrinum*, and P. *.chermesinum*, as shown in Table 2, Table 3 and Appendix A.

### 3.4. Molecular Data of Fungal Isolates Based on Multi-Locus Phylogenetic Analysis

For phylogenetic data, a standard fungal DNA barcode has been suggested and the quick identification of fungi has been made possible using ITS sequencing [52]. Although the ITS region has demonstrated high resolution power, which results in species discrimination between the most fungal taxa, the ITS region is only used for a few species-level identifications of *Aspergillus*, *Penicillium*, and *Talaromyces*. Another major drawback is its insufficient resolution in closely related species and species complexes involving cryptic species. Thus, the *RPB1*, *RPB2*, *BenA*, and *CaM* genes are presently used, while polyphasic taxonomy is increasingly used for species delimitation for fungal groups when ITS does not provide sufficient accuracy [52,53,54].

As shown in Figure 2, the phylogenetic tree had 19 isolates clustered in clades of *Aspergillus*, *Penicillium*, *Talaromyces*, and *Emericella*. Furthermore, the multi-locus phylogenetic analysis (Figure 2) showed that most representative isolates had moderate-to-high bootstrap values that were consistent with the BLAST results (Appendix A) and morphological-based classification (Figure 1).

Based on the concatenated ITS, *BenA*, *CaM*, and *RPB2* sequence data for phylogenetic analysis, the eight repetitive isolates of *A. niger* (B3-01, B8-02, B9-03, B16-04, G17-01, K6-03, K10-02, and M5-03) formed a monophyletic group and were phylogenetically related to *A. niger* CBS 554.65, as shown by the strong bootstrap support values (99% for MP and 100% for ML) and PP (1.00) at the nodes. As a result, eight repetitive isolates based on molecular evidence were correlated to the morphological data and confirmed as *A. niger*. The finding of this investigation showed the presence of *A. niger*, which was in accordance with other studies that reported this fungus in commercialized cereals [55,56].

*Aspergillus* sp. B1-01 grouped with *A. fumigatus* ATHUM 5013 in a strongly supported lineage, with bootstrap support values of MP = 100%, ML = 100%, and PP = 1.00; consequently, this isolate was identified as *A. fumigatus* B1-01. The phylogenetic placement of *Aspergillus* sp. G11-01 and *A. tubigensis* CBS 134.48 grouped them together (MP = 100, ML = 100, and PP = 1.00), with this information confirming G11-01 as *A. tubigensis* B1-01. In addition, both *A. tubigensis* CBS 134.48 and *A. tubigensis* G11-01 closely formed a sister group to a group of *A. niger* (B3-01, B8-02, B9-03, B16-04, G17-01, K6-03, K10-02, M5-03, ATHUM5044, and ATHUM2539) and *A. awamori* ATHUM 5181. Interestingly, our findings revealed that *A. fumigatus*, *A. niger*, and *A. tubingensis* were not only fungal contaminants in reported rice samples, but also enzyme-producing fungi belonging to the “Nigri section” in this case [57,58,59,60].

In the tree, *Penicillium* sp. G11-05 was placed with *P. chermesinum* CBS 231.81 (MP = 100, ML = 100, and PP = 1.00), while *Penicillium* sp. G11-02 was closely related to *P. citrinum* CBS 139.45 (MP = 100, ML = 100, and PP = 1.00), Thus, isolates G11-05 and G11-02 were identified as *P. chermesinum* G11-05 and *P. citrinum* G11-02, respectively. For *P. citrinum*, the present results indicated the occurrence of *P. citrinum*, which was consistent with other studies that showed the presence of *P. citrinum* in rice products [61,62,63].

Morphologically, *Penicillium* spp. (M1-01, M15-04, W2-03, and W3-04) formed a monophyletic group that was phylogenetically related to *T. pinophilus* CBS 631.66, as supported by the bootstrap support values (100% for MP and 97% for ML) and PP (1.00). In this case, although the data were inconsistent with the morphological results, we classified them as *T. pinophilus* (M1-01, M15-04, W2-03, and W3-04), based on the BLAST results and phylogenetic data. Isolate B8-03 was in a terminal taxon of *Talaromyces* and was closely related to *T. purpureogenus* strain CBS 286.36; thus, it was identified as *T. purpureogenus* B8-03. Based on the morphology, K1-02 and B5-02 were tentatively classified as the genus *Penicillium*; however, the sequence-based data revealed a match with *T. radicus* and *T. islandicus*. In the tree, K1-02 and B5-02 were clustered together (MP = 98, ML = 99, and PP = 1.00) and also grouped with *T. radicus* CBS 100489 and *T. islandicus* K1-02 to form a basal clade. Hence, we identified them as *T. radicus* B5-02 and *T. islandicus* K1-02, based on the BLAST results and morphological data. Notably, in the present study, for the first time the genus *Talaromyces spp* was clearly identified [64,65] in rice samples and confirmed using molecular techniques at the species level.

In summary, based on the concatenated phylogenetic methods (ITS, *CaM*, *BenA*, and *RPB2*) of the 19 fungal isolates obtained from the 40 rice samples, the molecular data and the BLAST results were in agreement with their morphological phenotypes. The two most frequently detected species were *A. niger* (eight isolates), and *T. pinophilus* (four isolates), followed by one isolate each of *A. fumigatus*, *A. tubingensis*, *T. radicus*, *T. purpureogenum*, *T. islandicus*, *P. citrinum*, and *P. chermesinum. Talaromyces* has not been reported in rice samples [61] and the present study was the first to show the presence of this fungal genus in rice samples using molecular techniques.

### 3.5. Mycotoxin Analysis

Different fungal strains cultured on PDA were analyzed for aflatoxins and OTA production using LC-MS/MS. Our data showed that one taxon of *T. pinophilus* W3-04, grown on PDA at 25 °C, produced OTA (Figure 3). In contrast, in the corresponding rice sample, OTA could not be detected. Surprisingly, the LC-MS/MS multi-toxin analysis identified in one of the rice samples the presence of BEA (Figure 4), a typical *Fusarium* toxin. Whether this unexpected finding relates to botanical impurities (cereals or other seeds) in the examined rice sample remains to be elucidated. However, BEA is considered as an emerging mycotoxin that has been recently detected in cereals and grains worldwide [66].

## 4. Discussion and Conclusions

Filamentous fungi (often described as molds in this context) and mycotoxins have become the most frequently measured food and feed contaminants at the global level, with the prevalence of fungal invasion still increasing as a reflection of global climate change. Of particular concern is the fungal and mycotoxin contamination of staple foods, such as grains including rice, which is the major dietary product in the Asia-Pacific region. Hence, the present study focused of fungal contamination and the presence of mycotoxins in commercial samples of rice taken at random from supermarkets in Bangkok, Thailand.

The results showed that *Aspergillus*, *Talaromyces*, and *Penicillium* were the three dominant fungal genera found in the commercial rice samples. All three genera are filamentous fungi belonging to the order Eurotiales and are known to be common contaminants of rice samples [67]. Furthermore, these three fungal genera are known to produce mycotoxins [68].

Several molecular markers were determined to provide greater insight into fungal contamination at the species level. Based on the concatenated phylogenetic methods (ITS, *CaM*, *BenA*, and *RPB2*) of the 19 fungal isolates obtained from the 40 rice samples, the molecular data and the BLAST results were in agreement with their morphological phenotypes. The two most frequently detected species were *A. niger* (eight isolates) and *T. pinophilus* (four isolates), followed by one isolate each of *A. fumigatus*, *A. tubingensis*, *T. radicus*, *T. purpureogenum*, *T. islandicus*, *P. citrinum*, and *P. chermesinum.* To date, *Talaromyces* has not been reported in rice samples [65,68] and the present study was one of the first to show the presence of this fungal genus in rice samples using molecular techniques. Some of the fungi detected in our investigation were comparable to those found in Nigeria, where the most common *Aspergillus* and *Penicillium* occurrences were *A. flavus* and *A. fumigatus* in rice samples and other food commodities [69]. Both *A. flavus* and *A. fumigatus* have been isolated from a variety of food materials across the world [70,71,72] and they are well adapted to rice grains as a nutrient source. Both these fungal species are known mycotoxin producers, including the production of aflatoxins by *A. flavus*. AFB1 is a regulated mycotoxin in most countries globally due to its epidemiologically confirmed involvement in the incidence of human hepatic carcinomas. *Aspergillus fumigatus* is a prominent producer of gliotoxins. The health hazard associated with A. fumigatus is not only related to the gliotoxin production but more importantly to shedding of a large number of micro-conidia, contaminating the environment. Occupational exposure or simple exposure in a household environment can cause serious allergic responses, aspergilloma, and invasive aspergillosis [73].

In our study, the marketed rice samples showed a high prevalence of *Aspergillus*, *Penicillium*, and *Talaromyces,* but no occurrences of *Cladosporium* and *Alternaria*. *A. niger* (eight isolates) was the most prominent species and our results supported other findings [74] reporting that *A. niger* was one of the most prevalent contaminants in commercially milled rice samples [74].

Comparing our results with other studies, *P. citrinum*, *P. islandicum*, and *P. verrucosum* were reported by Park et al. [19], showing that *P. citrinum* was the most detected species in polished rice, followed by *A. candidus*, *Altenaria* spp., and *A. versicolor*. In addition, they stated that the polished rice was contaminated with *Aspergillus* and *Fusarium* species, including *A. niger*, *A. flavus*, and *A. ochraceus*, with *Fusarium proliferatum* being the most prevalent species, followed by *F. oxysporum*, *F. graminearum*, and *F. semitectum* [18]. Another study identified *Eurotium*, *Muscor*, and *Rhizopus* and low frequencies of *Chaetomium* spp., *Cladosporium herbarum*, *Emericella nidulands*, *Geotrichum candidium*, *Phoma* spp., *Scopulariopsis* spp., and *Trichoderma* spp. [74]. The large variety of fungal species found in rice is likely related to the different geographical regions with different production and processing (transport and storage) techniques.

The regular presence of fungal contamination of rice samples has raised concerns about the potential health risks due to the presence of mycotoxins in marketed rice and rice products. Commonly, commercial samples of rice contaminated with AFB1 were related to an increased prevalence of hepatic cancers in humans. Subsequently, most countries have set maximum permissible levels for AFB1 and/or total aflatoxins (including AFB1, AFB2, AFG1, and AFG2) in foods, particularly for staple foods such as rice. However, these statutory limits vary between countries and are often related to the (traditional) food consumption pattern. In a recent review, it was reported that only one sample of AFB1 was found to exceed the Thai statutory limits [75]. This low incidence of contamination was also confirmed in the present study, as none of the investigated samples contained detectable AFB1. In contrast to AFB1, which is regulated in most countries, OTA is less commonly regulated, although both the WHO (JECFA) and the European Union have published recommendations for a maximum tolerable intake as a precautionary measure in consideration of the experimental evidence that OTA is involved in renal diseases and the incidence and progression of renal cancers and related urinary tract tumors. In Thailand, there is no regulation of OTA contamination at the national level. Our finding that OTA-producing fungal species might also occur on rice may stimulate further investigations on the prevalence of the OTA contamination of rice samples and discussion to endorse the statutory limits already defined by international organizations.

While formalized risk assessment by international agencies has focused initially on cancer as the most severe health hazard, currently, other adverse effects on humans and animals are being increasingly considered in overall risk characterization. Such effects include apparent food allergies (due to residual fungal material) and other adverse effects on the immune system that decrease the overall resilience to infectious disease in humans and animals. In this context, various additional mycotoxins are now denoted as emerging toxins, due to adverse findings related to human and animal health, and their occurrence in prominent food commodities, such as cereal grains and rice. BEA, measured in one of the rice samples investigated, is such an emerging toxin. While experimental data have identified numerous adverse effects [76] and various studies are in progress, no epidemiological data on humans are available.

Recent advances in analytical techniques allow the simultaneous monitoring of multiple mycotoxins. This is of relevance, as exposure to multiple mycotoxins can result in lower tolerance and an increased risk of adverse health effects. Therefore, it is recommended (in light of the occupational and consumer health risk associated with exposure to multiple mycotoxins) to monitor major food commodities regularly in all geographical regions.

In conclusion, the examination of 40 commercial rice sample collected in Bangkok, Thailand and the morphological and molecular analyses of 16,000 individual rice grains identified nine different fungal strains: *A*. *niger*, *A. fumigatus*, *A. tubingensis*, *Talaromyces pinophilus*, *T. radicus*, *T. purpureogenum*, *T. islandicus*, *P. chermesinum*, and *P. citrinum*. Most of these fungal species are known as mycotoxin producers. The potential for OTA toxin production was confirmed under experimental conditions for one of the strains of *T. pinophilus*, with this species being identified for the first time on rice samples. However, after applying a sensitive and validated LC-MS/MS multi-toxin analytical method, only in one sample was the mycotoxin BEA found, while the concentrations of all other mycotoxins remained below their LOQ levels. This also applied to AFB1, the only regulated mycotoxin in Thailand.

## Figures and Tables

**Figure 1 foods-12-00487-f001:**
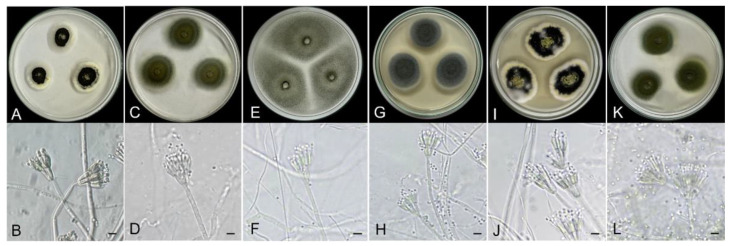
Morphological description on PDA at age 7 days and microscopic characteristics with conidial heads: (**A**,**B**) *Penicillium chermesinum* G11-05; (**C**,**D**) *P. citrinum* G11-02; (**E**,**F**) *Talaromyces islandicus* K1-02; (**G**,**H**). *T. pinophilus W3-04;* (**I**,**J**) *T. purpureogenum* B8-03; (**K**,**L**). *T. radicus* B5-02. Scale bars = 10 µm.

**Figure 2 foods-12-00487-f002:**
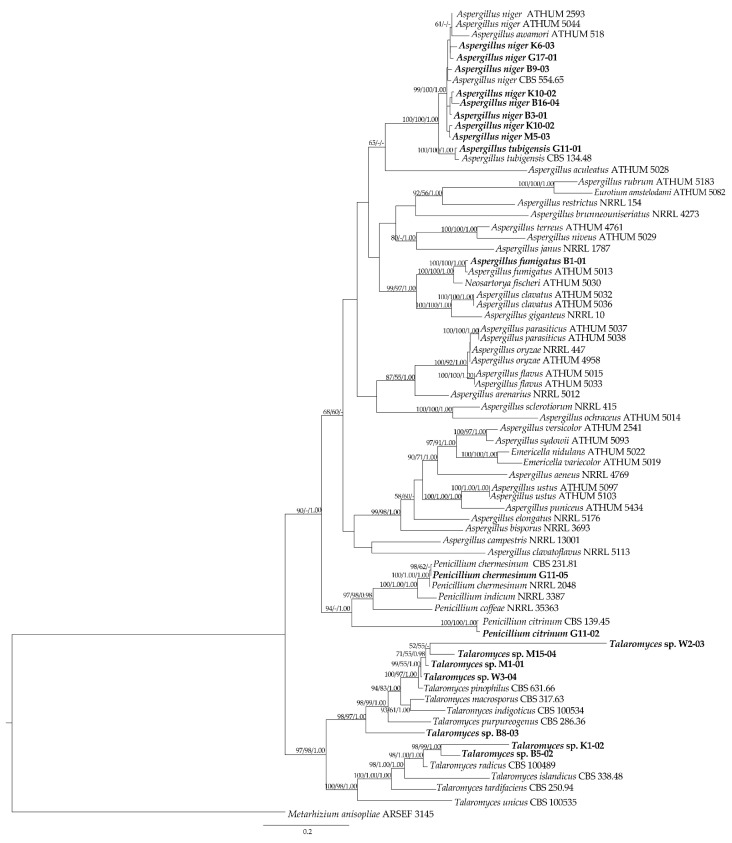
Maximum likelihood tree of 19 fungal isolates and allied species inferred from combined ITS, *BenA*, *CaM*, and *RPB2* dataset. Bootstrap support values for maximum likelihood (BSML, left), maximum parsimony (BSMP, middle) equal to or greater than 50% and Bayesian posterior probabilities (BPP, right) greater than 0.95 are indicated at the nodes. The 19 fungal isolates in the present study are indicated in bold letters. The tree is rooted with *M. anisopliae* (AF218207, AY995134, and DQ522453).

**Figure 3 foods-12-00487-f003:**
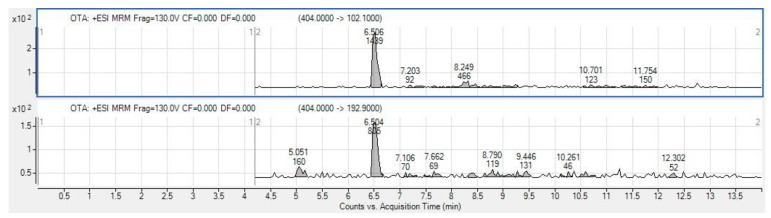
OTA extracted ion chromatograms of extracted fungal culture positive sample.

**Figure 4 foods-12-00487-f004:**
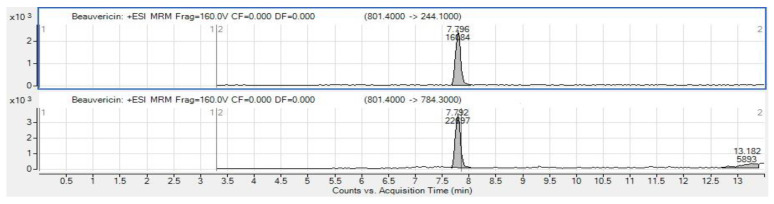
BEA extracted ion chromatograms of extracted rice positive sample.

**Table 1 foods-12-00487-t001:** Fungal molecular markers, primer names, direction, and amplification profile.

Molecular Marker	Primer	Direction	Reference	PCR Amplification		
				Denature	Repeat Step	Extension
Internal transcribed spacer (*ITS*)	ITS1	Forward	[32,33]	94 °C (2 min)	35 cycles, 94 °C (60 s), 55 °C (60 s), 72 °C (2 min)	72 °C (10 min)
	ITS5					
	ITS4	Reverse				
β-Tubulin (*BenA*)	Bt2a	Forward	[33]	94 °C (3 min)	35 cycles, 94 °C (60 s), 57 °C (60 s), 72 °C (2 min)	72 °C (10 min)
	Bt2b	Reverse				
Calmodulin (*CaM*)	cmd5	Forward	[35,36]	94 °C (10 min)	30 cycles, 94 °C (50 s), 55 °C (55 s), 72 °C (1 min)	72 °C (7 min)
	cmd6	Reverse				
RNA polymerase II second largest subunit (*RPB2*)	5F2	Forward	[34]	94 °C (2 min)	35 cycles, 94 °C (1 min), 55 °C (1 min), 72 °C (2 min)	72 °C (10 min)
	7cR	Reverse				

DNA sequencing was performed and the DNA sequences were compared with the database in GenBank (National Center for Biotechnology Information; NCBI; https://www.ncbi.nlm.nih.gov; accessed on 21 September 2022) using Standard Nucleotide BLAST (https://blast.ncbi.nlm.nih.gov/Blast.cgi, accessed on 12 August 2022) [39].

**Table 2 foods-12-00487-t002:** Prevalence of fungal contamination in 5 types of rice commercially available in Thailand determined based on blotting paper technique of individual rice grains.

Variety of Rice Sample	Number of Rice Samples	Total Tested Rice Seeds	Number of Contaminated Rice Seeds	Fungal Contamination (%)
Rice berry	13	5200	1291	24.83
Red jasmine rice	8	3200	1475	46.09
Brown rice	7	2800	405	14.18
Germinated brown rice	6	2400	1037	42.96
White rice	6	2400	77	3.21
Total	40	16,000	4285	26.78

**Table 3 foods-12-00487-t003:** Enumeration of grains contaminated with fungi on rice grains using agar plate method.

Variety of Rice Sample	Number of Rice Samples	Number of Fungal Isolates	Pathogen Incidence	
			*Aspergillus* spp.	*Penicillium* spp./ *Talaromyces* spp.
Rice berry	13	34	32	2
Red jasmine rice	8	20	14	6
Brown rice	7	12	9	3
Germinated brown rice	6	14	12	2
White rice	6	10	72	18
Total	40	90	72	18

## Data Availability

All newly generated sequences have been deposited in GenBank.

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
