# Peer review of "Storage Fungi and Mycotoxins Associated with Rice Samples Commercialized in Thailand"

_foods, 2023, doi:10.3390/foods12030487_

Round 1

Reviewer 1 Report

The presented study is interesting and the manuscript is well-written. However, the manuscript presents many gaps to be added and answered.

Comments:

Line 98: Add the number of samples for each rice sample between parentheses.

Line 102: Describe better the sample storage, including sample mass, package kind, whether the rice seeds were processed or not, and time of storage since the harvest.

Line 325: Figure 2 is impossible to read, please improve it.

Line 333: The results were not shown. The authors did not provide a quantitative result, and the figures of merits of the developed methodology had been not presented.  

Line 400: Provide maximum residues limit values according to Thailand regulation agencies or international agencies.

General comments: There is no information about MS results, like M/Z for each mycotoxin as well as monitoring ions. The authors did not provide any information about the analytical validation method. It will be adequate to compare the developed methodology with others. Include more discussion about mycotoxins content, the authors did promise this in the title of the submitted manuscript and did not provide it in the manuscript. The conclusion needs to be rewritten, there is poor relation with the manuscript's aim.

Author Response

Dear Reviewers;

We are pleased and grateful to the reviewers recognizing that the topic of the manuscript matches with the scope of the Foods. The authors would like to thank you for your time and feedback on our manuscript and provide the following responses addressing your comments.

Reviewer #1

The presented study is interesting and the manuscript is well-written. However, the manuscript presents many gaps to be added and answered.

Response: I appreciate very much for your valuable time for review. The comments from you are very useful for manuscript improvement.

Comments:

Line 98: Add the number of samples for each rice sample between parentheses.

Response: The number of samples was added for each rice sample between parentheses in the revised manuscript. (L107-108)

Line 102: Describe better the sample storage, including sample mass, package kind, whether the rice seeds were processed or not, and time of storage since the harvest.

Response: The information of rice samples was added in the revised manuscript. (L106, 108-109)

Line 325: Figure 2 is impossible to read, please improve it.

Response: The figure 2 was amended in the revised manuscript following the reviewer’s suggestion. (L339)

Line 333: The results were not shown. The authors did not provide a quantitative result, and the figures of merits of the developed methodology had been not presented.

Response: Accordingly, the information of quantitative results of mycotoxin production was added in the revised manuscript. (Figure 3 & 4; Supplementary 3-5)

Line 400: Provide maximum residues limit values according to Thailand regulation agencies or international agencies.

Response: The information was added in the revised manuscript (L432-440)

General comments: There is no information about MS results, like M/Z for each mycotoxin as well as monitoring ions. The authors did not provide any information about the analytical validation method. It will be adequate to compare the developed methodology with others. Include more discussion about mycotoxins content, the authors did promise this in the title of the submitted manuscript and did not provide it in the manuscript. The conclusion needs to be rewritten, there is poor relation with the manuscript's aim.

Response: The information about MS results and method validation were addressed in the revised manuscript (Supplementary 3-5). Together with this, the conclusion was rewritten as the reviewer’s recommendation. (L459-469)

Reviewer 2 Report

From the beginning, I emphasize that the 80 bibliographic references must be written according to the norms of the Journals of the publishing concern MDPI. The introduction describes the current state by using only 23 references out of 80 and of which a small number are current. In the experimental part, the most appropriate methods and analysis techniques are used. The experimental data are well developed, the two tables and respectively the two figures are well presented. The interpretation of the experimental data is done professionally. The conclusions are too brief, there are still new data obtained by the authors, which must be presented briefly. The bibliographic references will be redone in their entirety.

 Author Response

Dear Reviewers;

We are pleased and grateful to the reviewers recognizing that the topic of the manuscript matches with the scope of the Foods. The authors would like to thank you for your time and feedback on our manuscript and provide the following responses addressing your comments.

Reviewer #2

From the beginning, I emphasize that the 80 bibliographic references must be written according to the norms of the Journals of the publishing concern MDPI. The introduction describes the current state by using only 23 references out of 80 and of which a small number are current. In the experimental part, the most appropriate methods and analysis techniques are used. The experimental data are well developed, the two tables and respectively the two figures are well presented. The interpretation of the experimental data is done professionally. The conclusions are too brief, there are still new data obtained by the authors, which must be presented briefly. The bibliographic references will be redone in their entirety.

Response: I appreciate very much for your valuable time for review. The comments from you are very useful for manuscript improvement.

Accordingly, the conclusion was modified in the revised manuscript (L459-469). Together with this, bibliographic references were redone as the reviewer’ s suggestion. (Reference section)

Reviewer 3 Report

The manuscript foods-2129076 focused on the examination of different fungal species isolated from rice using conventional culture techniques and different molecular and phylogenic approaches. Mycotoxin production was also evaluated using the LC-MS/MS technique. In general, this study provides an insight into fungi currently present in five varieties of rice commercialized in Bangkok. From this point of view, the document is of interest to the scientific community.

I am attaching further remarks that may be of use to the authors.

L19-23. Please remove these lines.

L24-26. Mycotoxin production is missing…

L34. 8 isolates

L38. These finding provide a valuable guidance… Better written as: These preliminary findings provide…

L40-42. You need to insert results of the mycotoxin analysis in the abstract.

L41. ochratoxin A (OTA)

L53. 32.2 million metric tons

L54-57. The meaning of the sentence is not clear.

L58-59. by viable fungi… what does the word “viable” mean?

L62-64. Please use proper citation(s) to support this information.

L73. stable during proof processing… what does the word “proof” mean?

L75. aflatoxin B1 (AFB1)…

L78. (Class 1A)?

L78-79. Next to Aflatoxins, Ochratoxin A (OTA) has been identified a potential carcinogen.. Better written as: Ochratoxin A (OTA) has been considered as potential human carcinogen.

L83-87. The meaning of the sentence is not clear.

L87-89. Not clear what this means.

L95. LC-MS/MS

L98. 40 samples… in packaged form or loose direct to the consumer?

L100-101. Different brand samples … Six brands of rice in the Bangkok market are not enough as samples to cover the varieties. And plus, grain quality is not just dependent on the variety of rice, but quality also depends on the crop production environment, harvesting, processing, and so forth… The product information of the samples should be provided.

L103. The seed samples of rice… There is a great difference between seed and grain! Please use grain instead of seed.

L106. Distilled water… sterile distilled water?

L126. Why did the authors choose this incubation temperature?

L159. Mycotoxin production… The term “mycotoxin” refers to total aflatoxins, OTA, total fumonisins, zearalenone, deoxynivalenol, citrinin, nivalenol, diacetoxyscripenol, T-2 toxin, beauvericin, and alternariol ?

L162. Muñoz et al. evaluated the mycotoxin production of Aspergillus and Penicillium species on coffee- and wheat-based media. Did the authors evaluate the behavior of toxigenic fungal species when cultured on food based media?

L163. at day 7 of incubation… Why did the authors choose this time-window? Mycotoxin production clearly depends upon time of incubation, fungal species, and medium composition!

L165-166. min instead of “mins”.

L166. Centrifugation conditions?

L175-176. Then, 2.0 g of  MgSO4

L178. Please, never begin a sentence with a numeral.

L179 and L287-288. PSA, MP, ML, PP? The first time you use an abbreviation in the text, present both the spelt-out version and the short form.

L178 and 180. 1,968 × g

L183. Injected or passed through?

L185. Perhaps the detailed description of the LC-MS/MS methodology is not necessary as long as a method reference or a previous work in the authors’ laboratory is provided. This information could be moved to a supplementary file with a sample chromatogram. However, the limits of detection (LOD) and/or quantification (LOQ) obtained by the technique should be specified in this section.

L218. Statistical analysis section is missing!

L218. Results or Results and Discussion? Please improve the results section with clear findings.

L226-227. All the 4,285 infected rice grains were categorized based on their morphological characteristics?

L228.. of 5 rice seeds…?

L235-237. I suggest removing any discussion of the obtained results from the results section.

L235. Table 2 or Table 3?

L235-236. Republic of Korea

L249. Supplementary 1?

L258-262. The sentence is somewhat confusing.

L270-274. The meaning of the sentence is not clear.

L279. The quality of the figure is very poor. I strongly recommend the author embellish the figure.

L295. Aspergillus sp. G11-01

L308. …specie

L323. …molecular techniques at the species level.

L331. Data on mycotoxin analysis is missing! Could the authors provide the mycotoxin content of all tested samples?

L334. Please specify the content of OTA!

L336. Please specify the content of BEA!

NOTE: It is very important to distinguish between primary mycotoxin production in the field —pre-harvest and harvest— and production of mycotoxins during storage, which is the most frequent. Moreover, Authors should insert the moisture content of all evaluated samples in the corresponding table. Otherwise, the mycotoxin contents need to be expressed as μg/kg dry weight.

L341. To make the manuscript academically and scientifically sound, please combine the discussion with the results section.

L342-346. The sentence is somewhat confusing.

L349-350. …staple food such as cereals and grains… cereals and grains?

L364-366. The meaning of the sentence is not clear.

L415. I think you can end up the discussion section with more conclusions from your study.

Author Response

Dear Reviewers;

We are pleased and grateful to the reviewers recognizing that the topic of the manuscript matches with the scope of the Foods. The authors would like to thank you for your time and feedback on our manuscript and provide the following responses addressing your comments.

Reviewer #3

 The manuscript foods-2129076 focused on the examination of different fungal species isolated from rice using conventional culture techniques and different molecular and phylogenic approaches. Mycotoxin production was also evaluated using the LC-MS/MS technique. In general, this study provides an insight into fungi currently present in five varieties of rice commercialized in Bangkok. From this point of view, the document is of interest to the scientific community.

Response: I appreciate very much for your valuable time for review. The comments from you are very useful for manuscript improvement.

I am attaching further remarks that may be of use to the authors.

L19-23. Please remove these lines.

Response: These lines were removed from the revised manuscript as the reviewer’s recommendation.

L24-26. Mycotoxin production is missing…

Response: This information was added in the revised manuscript accordingly. (L21-23)

L34. 8 isolates

Response: Thank you very much. The word was amended as the reviewer’s comments. (L30)

L38. These finding provide a valuable guidance… Better written as: These preliminary findings provide…

Response: The sentence was amended as the reviewer’s comments. (L37)

L40-42. You need to insert results of the mycotoxin analysis in the abstract.

Response: The information of the mycotoxin analysis was added in the abstract of the revised manuscript as the reviewer’s recommendation. (L34-42)

L41. ochratoxin A (OTA)

Response: The word was amended in the revised manuscript. (L40)

L53. 32.2 million metric tons

Response: The word was added in the revised manuscript. (L53)

L54-57. The meaning of the sentence is not clear.

Response: The sentence was slightly amended in the revised manuscript. (L58-61)

L58-59. by viable fungi… what does the word “viable” mean?

Response: The sentence was modified in the revised manuscript accordingly. (L64-66)

L62-64. Please use proper citation(s) to support this information.

Response: Accordingly, the proper citation was amended in the revised manuscript. (L66-68)

L73. stable during proof processing… what does the word “proof” mean?

Response: Accordingly, the word was amended in the revised manuscript. (L78)

L75. aflatoxin B1 (AFB1)…

Response: The word was amended in the revised manuscript. (L82)

L78. (Class 1A)?

Response: Accordingly, the word was amended in the revised manuscript. (L84-85)

L78-79. Next to Aflatoxins, Ochratoxin A (OTA) has been identified a potential carcinogen. Better written as: Ochratoxin A (OTA) has been considered as potential human carcinogen.

Response: The sentence was amended in the revised manuscript. (L85)

L83-87. The meaning of the sentence is not clear.

Response: The sentence was modified in the revised manuscript accordingly. (L90-92)

L87-89. Not clear what this means. 

Response: The sentence was modified in the revised manuscript accordingly. (L92-96)

L95. LC-MS/MS

Response: The word was amended in the revised manuscript. (L102)

L98. 40 samples… in packaged form or loose direct to the consumer?

Response: The sentence was modified in the revised manuscript accordingly. (L106)

L100-101. Different brand samples … Six brands of rice in the Bangkok market are not enough as samples to cover the varieties. And plus, grain quality is not just dependent on the variety of rice, but quality also depends on the crop production environment, harvesting, processing, and so forth… The product information of the samples should be provided.

Response: The product information of rice samples was added in the revised manuscript accordingly. (L108-109)

L103. The seed samples of rice… There is a great difference between seed and grain! Please use grain instead of seed.

Response: The word was modified throughout the revised manuscript. (L112)

L106. Distilled water… sterile distilled water?

Response: The word was added in the revised manuscript accordingly. (L114)

L126. Why did the authors choose this incubation temperature?

Response: Authors thanks for your question. Together with this, we added the range of temperature for fungal incubation during 25-28 °C in agreement with the previous publication (Fredricks et al., 2005).  (L135)

L159. Mycotoxin production… The term “mycotoxin” refers to total aflatoxins, OTA, total fumonisins, zearalenone, deoxynivalenol, citrinin, nivalenol, diacetoxyscripenol, T-2 toxin, beauvericin, and alternariol ?

Response: Yes, the information was added in the revised manuscript accordingly. (L171-173)

L162. Muñoz et al. evaluated the mycotoxin production of Aspergillus and Penicillium species on coffee- and wheat-based media. Did the authors evaluate the behavior of toxigenic fungal species when cultured on food based media?

Response: Unfortunately, authors do not evaluate, however, it is interesting to evaluate in the future.

L163. at day 7 of incubation… Why did the authors choose this time-window? Mycotoxin production clearly depends upon time of incubation, fungal species, and medium composition!

Response: Authors have chosen at day 7 of incubation because it is the optimal time that fungi can produce spores completely.

L165-166. min instead of “mins”.

Response: The word was corrected in the revised manuscript. (L179-180)

L166. Centrifugation conditions?

Response: The information was added in the revised manuscript. (L180-181)

L175-176. Then, 2.0 g of MgSO4…

Response: The word was corrected in the revised manuscript. (L189-190)

L178. Please, never begin a sentence with a numeral.

Response: The sentence was modified in the revised manuscript. (L192-193)

L179 and L287-288. PSA, MP, ML, PP? The first time you use an abbreviation in the text, present both the spelt-out version and the short form.

Response: Accordingly, authors have revised following the reviewer’s suggestion.

 “Bayesian inference” was changed into “Bayesian posterior probabilities (PP)”, and additional sentence was added as “Bootstrapping for maximum parsimony (MP) and maximum likelihood (ML) was performed as following method of Raja et al. (2017) [32]”. (L163-166 and L194)

L178 and 180. 1,968 × g

Response: The information was added in the revised manuscript. (L192 and 195)

L183. Injected or passed through?

Response: The word was corrected in the revised manuscript accordingly. (L197)

L185. Perhaps the detailed description of the LC-MS/MS methodology is not necessary as long as a method reference or a previous work in the authors’ laboratory is provided. This information could be moved to a supplementary file with a sample chromatogram. However, the limits of detection (LOD) and/or quantification (LOQ) obtained by the technique should be specified in this section.

Response: The information of method validation was added in the revised manuscript as referee’s suggestion (L222-224 and Supplementary 3-5)

L218. Statistical analysis section is missing!

Response: For this section, authors do not use the statistical analysis among types of rice.

L218. Results or Results and Discussion? Please improve the results section with clear findings.

Response: Accordingly, authors attempt to improve the results clearly in the revised manuscript.

L226-227. All the 4,285 infected rice grains were categorized based on their morphological characteristics?

Response: Certainly, all the infected rice grains found were categorized based on their morphological characteristics but they were separated into group with the result on PDA agar. Then, we observed under microscopic.

L228 ... of 5 rice seeds…?

Response: The sentence was modified in the revised manuscript. (L244-245)

L235-237. I suggest removing any discussion of the obtained results from the results section.

Response: With high respect, authors attempt to explain the results just in line with the other results.

L235. Table 2 or Table 3?

Response: Authors appreciate very much. The word was amended in the revised manuscript. (L251)

L235-236. Republic of Korea

Response: The word was corrected in the revised manuscript. (L252)

L249. Supplementary 1?

Response: The information was shown in Supplementary 1.

L258-262. The sentence is somewhat confusing.

Response: Authors agree as the referee’s comments. The sentence was modified in the revised manuscript. (L274-277)

L270-274. The meaning of the sentence is not clear.

Response: Accordingly, the sentence was modified in the revised manuscript. (L285-289)

L279. The quality of the figure is very poor. I strongly recommend the author embellish the figure.

Response: The figure was amended in the revised manuscript in agreement with referee’s comments. (L339)

L295. Aspergillus sp. G11-01

Response: The word was amended in the revised manuscript. (L310)

L308. …specie

Response: The word was modified in the revised manuscript. (L323)

L323. …molecular techniques at the species level.

Response: The word was amended in the revised manuscript. (L338)

L331. Data on mycotoxin analysis is missing! Could the authors provide the mycotoxin content of all tested samples?

Response: Data on mycotoxin analysis was added in the supplementary 3-5.

L334. Please specify the content of OTA!

Response: Accordingly, the figure 3 was added to specify OTA analysis in the revised manuscript.

L336. Please specify the content of BEA!

Response: Accordingly, the figure 4 was added to specify OTA analysis in the revised manuscript.

NOTE: It is very important to distinguish between primary mycotoxin production in the field —pre-harvest and harvest— and production of mycotoxins during storage, which is the most frequent. Moreover, Authors should insert the moisture content of all evaluated samples in the corresponding table. Otherwise, the mycotoxin contents need to be expressed as μg/kg dry weight.

Response: In agreement with the referee’s comments, the mycotoxin levels were expressed as μg/kg in the revised manuscript.

L341. To make the manuscript academically and scientifically sound, please combine the discussion with the results section.

Response: Accordingly, we combined the discussion and conclusion section together to make academically and scientifically sound in the revised manuscript in agreement with the referee’s suggestion. (L371)

L342-346. The sentence is somewhat confusing.

Response: The sentence was modified in the revised manuscript (L372-375)

L349-350. …staple food such as cereals and grains… cereals and grains?

Response: The sentence was modified in the revised manuscript (L375-376)

L364-366. The meaning of the sentence is not clear.

Response: The sentence was modified in the revised manuscript (L346-353)

L415. I think you can end up the discussion section with more conclusions from your study.

Response: Accordingly, more conclusions from our study were added in the revised manuscript. (L459-469)

Reviewer 4 Report

The authors are clear in the study objectives and description of the methods. Overall, the paper does need editing for English grammar. The conclusions of the study would be more effective if presented in a table or figure.

The primary limitation of the manuscript is in the discussion explaining the significance of the results. What are the public health implications of the findings? How or why does geographic location matter? Expand the discussion regarding occpational health risks, as well as consumer health risks.   

Author Response

Dear Reviewers;

We are pleased and grateful to the reviewers recognizing that the topic of the manuscript matches with the scope of the Foods. The authors would like to thank you for your time and feedback on our manuscript and provide the following responses addressing your comments.

Reviewer #4

The authors are clear in the study objectives and description of the methods. Overall, the paper does need editing for English grammar. The conclusions of the study would be more effective if presented in a table or figure.

Response: Accordingly, the revised manuscript was edited for English grammar by the native speakers, and more conclusions were added in the revised manuscript. (L459-469)

The primary limitation of the manuscript is in the discussion explaining the significance of the results. What are the public health implications of the findings? How or why does geographic location matter? Expand the discussion regarding occupational health risks, as well as consumer health risks.  

Response: In agreement with the referee’s suggestion, the information of occupational and consumer health risk estimation was added in the revised manuscript. (L452-457)

Round 2

Reviewer 1 Report

Based on the addressed and added information, I recommend this publication. 

Reviewer 2 Report

The authors responded to the reviewers' comments and corrected the manuscript.

I agree with the publication of the manuscript in the corrected form.

Reviewer 3 Report

The document has been corrected according to my previous comments. I have no further suggestions. The manuscript could be published in its present form.